# A deep neural network model for multi-view human activity recognition

**Prasetia Utama Putra** [1]*, **Keisuke Shima**[2], **Koji Shimatani**[3]

**1** Graduate School of Engineering, Yokohama National University, Yokohama, Kanagawa, Japan, **2** Faculty of Engineering, Yokohama National University, Yokohama, Kanagawa, Japan, **3** Faculty of Health and Welfare, Prefectural University of Hiroshima, Hiroshima Prefecture, Hiroshima, Japan

* prasetia-putra-kt@ynu.jp

## Abstract

Multiple cameras are used to resolve occlusion problem that often occur in single-view human activity recognition. Based on the success of learning representation with deep neural networks (DNNs), recent works have proposed DNNs models to estimate human activity from multi-view inputs. However, currently available datasets are inadequate in training DNNs model to obtain high accuracy rate. Against such an issue, this study presents a DNNs model, trained by employing transfer learning and shared-weight techniques, to classify human activity from multiple cameras. The model comprised pre-trained convolutional neural networks (CNNs), attention layers, long short-term memory networks with residual learning (LSTMRes), and Softmax layers. The experimental results suggested that the proposed model could achieve a promising performance on challenging MVHAR datasets: IXMAS (97.27%) and i3DPost (96.87%). A competitive recognition rate was also observed in online classification.

**Data Availability Statement:** The data underlying this study are publicly available at the following links: IXMAS: https://www.epfl.ch/labs/cvlab/data/data-ixmas10/ i3dPost: http://kahlan.eps.surrey.ac.uk/i3dpost_action/ The authors confirm that they did not have any special access privileges and

## Introduction

Occlusion causes information loss and failure in single-view human activity recognition [1]. Previous researchers [2–7] have attempted to resolve this issue with the use of multiple cameras providing different angles of view [1, 6] and enabling 3D posture representation [8, 9]. Multi-view human action recognition (MVHAR) has a wide range of applications, including in systems for surveillance [4, 10] and human behavior monitoring [5, 11].

Current multi-view approaches comprise conventional computer vision (CV) or DNNs methods [12, 13]. The conventional methods require sophisticated features extraction to identify informative features from raw data [14–16]. The features extractor usually works independently from the classifier [17, 18]. Studies based on this approach focus either on the classifier or on feature engineering [19, 20].

Representing human action from multiple views is the major challenge in feature engineering studies for multi-view action recognition. Previous studies have encoded human movement as low-level representation such as histograms of gradient (HoG) [14], silhouettes [15], and optical flow [21] that were extracted from RGB images. Afterwards, they were used for

others would be able to access these data through the methods described here.

**Funding:** This work was supported by JSPS Kakenhi under Grant No. 26285212 (K.S) and 18H01041 (K.Sh). JSPS: https://www.jsps.go.jp/.

**Competing interests:** The authors have declared that no competing interests exist.

direct classification or transformed to higher-level features [9, 15, 16, 22]. Action recognition was performed either by fusing features from multiple views followed by classifier algorithms [8, 9] or by estimating actions in an individual view and then combining or voting for prediction scores [22]. Studies involving classifiers for multi-view techniques intended to improve recognition rates based on the efficient utilization of features from multiple cameras [17, 18].

Conversely, the DNNs approach combines feature extractors and classifiers into a single pipeline [23–25]. A DNNs model automatically discovers features representation and recognizes data patterns based on an end-to-end learning algorithm [26] that trains the model to extract informative features from the data for specific classes [27]. DNNs have outperformed conventional methods in many CV fields, such as object recognition [28] and video classification [29]. However, the performance of current DNN does not match that of state-of-the-art methods [23, 30], partly because of the difficulty of training DNN with the limited number of samples in MVHAR datasets [22, 31].

Previous works on DNNs in MVHAR have involved early [23] or late fusion to combine multiple inputs [30]. Early fusion resulted in a DNNs model with a modest number of parameters, which combined features from the early layer [23]. However, the combinations of features of early layer had high variance, making it prone to over-fitting. Meanwhile, late fusion combined features from multiple cameras by treating inputs individually with multiple models [25, 30]. Individual models in this approach may have fewer variant features but a high number of parameters, which consume more memory. Other DNNs approaches have attempted to solve the multi-view human action recognition by employing multimodal inputs [32–36], multi-task training [25], and cross-view learning [24, 37] algorithm.

This paper presents a novel DNNs model based on a shared-weight application for multi-view human action recognition, supporting late fusion with fewer parameters than the multi-model technique. The model involved the use of multi-view images as input to produce multiple hypotheses, and score-fusion was used to compute the final prediction. As the prior details of informative input among multiple views were unknown, the proposed model was trained to treat individual prediction scores equally with the arithmetic mean or weight the hypothesis with the geometric mean. The model applied an attention network to filter out uninformative features from the sequence of images.

The model comprises pre-trained CNNs, attention layers, RNNs, and Softmax layers. Exploration studies were conducted for structural optimization, and transfer learning was performed with pre-trained CNNs to prevent over-fitting in the training process. We compared the performance of the proposed model performance to that of the-state-of-art application on IXMAS [22] and i3DPost [31]. We conducted an online evaluation and comparative study with the single-view model to determine its efficiency in the actual situation.

The study can be summarized as follows:

- A novel DNNs model employing shared weight for MVHAR was proposed. Despite the model's fewer parameters, higher accuracy was observed from the application of shared-weight than from multi-model DNNs usage.

- The effects of applying residual shortcuts on LSTM and employing intermediate and last-layer pre-trained CNNs to extract features were investigated.

- Effectiveness of features and score fusion for the combination of information from multi-view inputs were compared.

- Evaluation of the model for single-view and multi-view inputs suggested an increased recognition rate with the latter.

- Evaluation of the model's performance during actual application in online classification indicated that longer image sequences produced higher recognition rates.

## Related work

In the last decade, conventional CV approaches have dominated the MVHAR field; they represented human body configuration using 2D, 3D, and 4D models. Methods using 2D models extracted silhouettes and optical flow from sequences of images for direct classification [9, 15] or transformation to higher-level features [1, 18, 38, 39]. High-level features such as silhouettes contour points and centers of mass [15] showed superiority over other methods employing 2D data [18] to encode movement in human action.

The conventional CV approaches with 3D/4D models required a sophisticated algorithm to extract informative features from RGB images [8, 9, 17, 21]. The existing approaches either directly concatenated all features from multiple views [8, 9, 22] or weighted multiple hypotheses [13, 21] from those inputs to discriminate the human activity. Pehlivan *et al.* [8] encoded sequences of silhouettes from multiple views as cylindrical shapes with different rotations, while Weinland *et al.* [22] extended motion history (MH) determined from a single view to a motion history volume that combined MH from multiple views. With a six-step feature extractor, Holte *et al.* [9] determined 4D spatio-temporal interest points and local descriptions of 3-D motion features from image sequences. Another study [40] combined local and global features with self-similarity matrix; the study did not require 3D model to represent subjects' activity from multiple views. These approaches classified human activity by feeding the combined features to a classifier algorithm.

In contrast with features-fusion, score-fusion involves separate treatment of input features, followed by a combination of hypotheses from all inputs using weighting functions. Previous works have involved score-fusion using the arithmetic mean [30], fixed weight operation [13], and a data-driven adaptive weight algorithm [21]. Arithmetic means assumed that prior knowledge of informative views was unknown; it treated all confidence scores equally [30]. Fixed weight operation involved learning to identify informative inputs from the data [13], while adaptive algorithm weighted the hypothesis with different masks during the inference [21].

In learning representation with DNNs, many researchers [23, 25, 30, 32, 34, 41] have proposed MVHAR methods based on CNNs. Kavi *et al.* [30] proposed a DNNs model compromising multiple models to handle multi-view inputs to determine human action. Multi-branch of CNN has also been used to perform view-specific action recognition from multiple views using view-specific classifier [25, 41]. Although these approaches exhibited promising performance, multiple DNNs models required a significant number of parameters to compromise the number of inputs. Accordingly, Khan *et al.* [32] employed a single pretrained CNNs to extract features from multi-view inputs for combination with hand-crafted features. Then, they performed features selection [42] to select robust features before classifying them. This technique allowed the proposed model to estimate human action from multiview with fewer parameters.

Recent studies have attempted to improve further recognition rate of DNNs in MVHAR by training the model with multiple modalities [21, 34, 36, 43–45], multi-task [24], and crossview [41] techniques. Multimodal approaches combine 2D-RGB images with higher-level inputs, such as optical flow [21, 34], depth information [43], radar sensors [45], and skeleton features [36, 44]; the model consisted multiple streams that proceeded with different type of modalities. Multi-task assumed the model could produce informative latent variables by

simultaneously learning different but related tasks: predicting human activities from multi-view inputs and the inputs' view-index [24]. Cross-view methods trained the model to predict how the activities videos seen from a different viewpoint [37, 41]; the methods used inputs from different views during training and test.

In previous work [23], we proposed a DNNs model for multi-view human action recognition employing a multiple DNNs models to handle multiple inputs. Evaluated with IXMAS dataset [22], the proposed model achieved comparable results. However, it often misclassified actions performed by hands. Using multi-model to process multi-view inputs also increased the number of parameters and computation complexity.

This study aimed to resolve the multi-model issue and investigate the performance of the DNNs model with features and score fusions in MVHAR. We shared a single CNNs block and LSTMRes block across multiple inputs and then fused multiple hypotheses produced by the model using score fusion to predict human activity. Previous researchers combined scores with weighting functions that determined parameter values during training [13] or inference [21]. In contrast, in the proposed model, it was assumed that there was no prior knowledge of informative views. Therefore, prediction scores from individual view inputs are treated equally with the arithmetic mean or weighted with the geometric mean during training and inference.

The proposed model employed RGB images as inputs and did not combine the latent variables with another modality. The work examined the proposed model's performance with the IXMAS [22], and i3DPost [31] datasets and evaluated its implementation in an online scenario.

## Material and methods

### Proposed DNNs model

The proposed DNNs model comprised a pre-trained CNNs, an attention layer, an RNNs layer, and a score-fusion layer [26] (see Fig 1), with multiple inputs and outputs representing multiple views and actions.

### Pre-trained CNNs

Pre-trained CNNs were used to extract spatial information from RGB images in the proposed model. The pre-trained models' intermediate or final layers' output were used to extract features from sequences of images. The output comprised a feature map $f$ of shape $H \times W \times C$, where $H$, $W$, and $C$ are height, width, and channel, respectively. Hence, the feature vector for the $T$ time-step was

$$F = [f_1, ..., f_T], \quad \text{with } f_t \in \mathbb{R}^G \text{ and } G = H \times W \times C \qquad (1)$$

This study involved an examination of the pre-trained models VGG-19 and VGG-16 [46] comprising five blocks with different numbers of CNNs. VGG-16 comprised 12 stacked CNNs: two in 1st, 2nd, and 3d blocks, and three in 4th and 5th blocks. While VGG-19 had an extra CNN in the 4th and 5th blocks, making 14 stacked CNNs in total. This paper refers to $I$-th CNN in $N$-th block as `blockN_convI`.

### Attention layer

Since it was assumed that significant transformation occurred only in certain parts of image sequences when subjects performed actions, the proposed model filtered out out uninformative features by employing an attention layer [47] that weighted important features with higher probability and the others with lower probability.

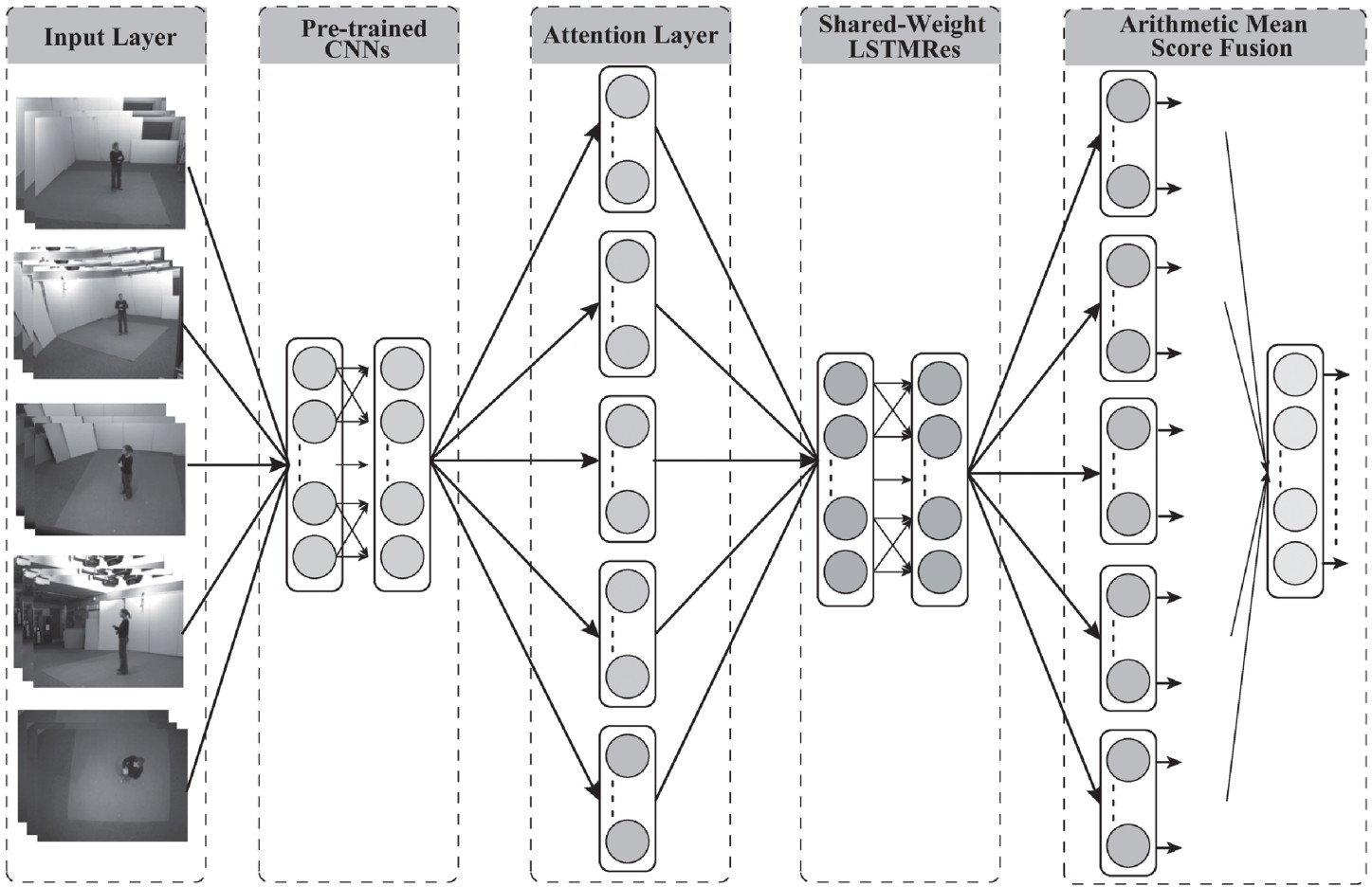

**Fig 1. Architecture of the proposed model.** Pre-trained CNNs and LSTMRes were shared across inputs.

Given the feature vector $F$ of shape $T \times G$, the attention mask was computed by averaging attention scores over $G$. The first step to determine relevant features was to estimate attention probability at each time step for the $G$ dimension. For the feature map at the $t$-th time step $f_t$, attention probability was given by

$$s_t = g_{\mathrm{att}}(f_t; \theta_t) \qquad (2)$$

$$\alpha_t = \mathrm{softmax}(s_t) \qquad (3)$$

where $g_{\mathrm{att}}$ was an attention network with weight $\theta_t$, and $s_t$ was the attention score map for the feature map. The attention score $\alpha_t$ was the probability produced from Softmax function incorporating the subject of interest with a higher probability than the rest. Dense, convolutional, and RNNs layers can be used as attention networks [48]; the proposed model employed a dense layer for the attention network because, in the preliminary experiment, we found CNNs and RNNs caused over-fitting.

After computing attention probability at each time step, the relevant features were calculated using

$$\hat{f}_t = f_t \odot \alpha_t \qquad (4)$$

where $\odot$ represents the element-wise operator or the Hadamard product [49] weighting features extracted from pre-trained CNNs with $\alpha_t$.

## Residual learning in LSTM

A long short-term memory (LSTM) architecture [50] was proposed to solve the problem of vanishing and exploding gradients associated with conventional recurrent neural networks (RNNs) [26]. The architecture, however, still can suffer from degradation problems caused by deeper neural network structure [51]. Residual learning was proposed to tackle this issue by introducing a shortcut connection from the earlier to the later layers that helps the earlier layer get a-"fresh"-gradient from the latter one during backpropagation [52].

In contrast to the highway network approach [53], residual learning formulation [52] involved an identity shortcut to ensure ongoing learning. Residual function $H(z_i)$ could be expressed as:

$$H(z_i) = F_i(z_i, W_i) + W_s z_{i-m} \tag{5}$$

where $F(z_i, W_i)$ and $z_{i-m}$ represent the original mapping and output from the earlier layer, respectively and $W_s$ was a linear projection that was used when the dimension between $F(z, W_i)$ and $z_{i-m}$ was unequal, as realized via linear mapping.

In LSTM, residual mapping could be accomplished by introducing a shortcut connection to the adjacent layer, from layer $t$ to $t + 1$ [54] (Eq 6), or by establishing a connection to the memory cell [23] (Eq 7, implementation: Fig 2).

$$h_t = o_t \odot tanh(C_t) + h_{t-1} \tag{6}$$

$$h_t = o_t \odot tanh(C_t + W_s x_t) \tag{7}$$

Here, $o_t$, $C_t$, $h_t$ represent the output gate, memory cell, and hidden units, respectively.

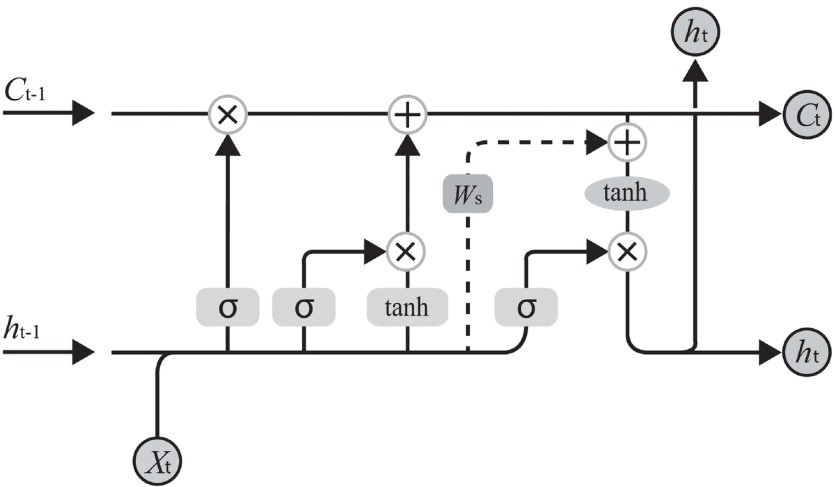

**: Neural network layer ◯: Point wise**

**Fig 2. Architecture of LSTM with residual learning.** Implementation of residual learning in LSTM with shortcut connection after forgetting of old information and addition of the new information. The dotted line shows shortcut connection.

## Shared weight LSTMRes

The model in our previous work used Multiple Sequence LSTMRes (MSLSTMRes) to decode temporal deformation changes in features from multiple cameras [23]. MSLSTMRes experimentally outperformed baseline topology at the expense of computational time.

To address that issue, the proposed model used shared weights of pre-trained CNNs and stacked LSTMRes (comprising two LSTMRes with 512 units) across inputs from all cameras. Previous work applied a shared hidden-layer network to find similarities in speech and text [55]. This work employed a shared-weight model to learn transformation and similarity among features from multiple cameras, enabling late fusion using only a single model.

## Score fusion

Arithmetic or geometric means are used to combine prediction scores from Softmax layers. With the former, scores from all cameras were treated as a mixture, while the latter allows one prediction result from a single camera to veto other outcomes. The proposed model calculated final prediction scores using arithmetic mean (Eq 8) or geometric mean (Eq 9).

Here, $y_{ac}$ represents the probability score of an action $a$ from camera $c$. $M$ and $N$ are respectively the total number of actions and cameras.

$$Y_a = \frac{\sum_c^N y_{ac}}{N} \tag{8}$$

$$Y_a = \frac{\sqrt[N]{\prod_c^N y_{ac}}}{\sum_a^M \sqrt[N]{\prod_c^N y_{ac}}} \tag{9}$$

## Datasets and evaluation metrics

The IXMAS dataset [22] is a benchmark in MVHAR algorithm evaluation that comprises videos of 12 subjects performing 13 actions: watch checking, arms crossing, head-scratching, sitting, getting up, turning around, walking, waving, punching, kicking, pointing, picking something up, and throwing. Videos were recorded using five cameras at 23 fps. Subjects performed each action three times with free positioning and orientation.

The i3DPost dataset [31] was recorded using eight synchronized cameras with a resolution of 1920x1080 and 25Hz progressive scan. The eight subjects performed 12 actions (walking, running, jumping, bending, waving, jumping in place, sitting-standing, running-falling, walking-sitting, running-jumping-walking, hand-shaking, and pulling), creating 96 multi-view videos of human activity.

The proposed model's performance was evaluated with categorical cross-entropy loss, classification accuracy, and F1-score metrics. The accuracy rate was computed by averaging top-1 accuracy for given data, while F1-score was the average F1-score for all classes. $p$-value was computed using Student $t$ test [56].

## Pre-processing and learning

To reduce distortion in images and ensure the features were on the same scale, RGB-normalization and feature standardization were performed in pre-processing. The mean and standard deviations were computed individually for each dataset to standardize the value of images. Gamma correction was applied to images of IXMAS dataset; the gamma value was 1.5.

In the experiments, the proposed model was trained with three scenarios (Table 1):

**Table 1. Scenarios used in the experiments.**

| Scenario | lr | D, RD | MB | Decay | Image Size |
|----------|------|----------|----|-------|------------|
| I | 1.E-4 | 0.5, 0.3 | 15 | No | 73x73 |
| II | 1.E-4 | 0.5, 0.3 | 7 | Yes | 128x128 |
| III | 9.5E-5 | 0.1, 0.1 | 7 | Yes | 128x128 |

lr, D, RD, and MB represent the learning rate, input and recurrent dropout, and mini batch, respectively. The-"Decay"-column shows whether learning rate decay was used. The decay factor was 0.31 with patience of 20 iterations. The mini-batch value depended on image size.

1. scenario I: LSTMRes evaluation,

2. scenario II: evaluation of the pre-trained model, MSLTMRes and score fusion, and implementation of online classification,

3. scenario III: investigation of multi-view inputs and a comparison of the proposed model with state-of-the-art methods.

In all scenarios, backpropagation with RMSProp optimizer [57] was used.

Glorot uniform [58] and orthogonal [59] initializers were used to initialize the parameter values of kernel and recurrent weights, respectively; the bias values were initialized to be zero. As this study used pre-trained CNNs in all experiments, it did not apply parameters initialization to CNNs.

Evaluation in scenario II and III involved one-leave-subject cross-validation, while scenario I used train-test evaluation. We also applied early stopping during training in scenario II and III.

Action image sequences were trimmed to 22 frames for the scenario I and 20 frames for the other scenarios; experimental results showed that using 20 frames resulted in higher accuracy of the proposed model. Data augmentation was performed in scenario III by sub-sampling a frame sequence with different frequencies to prevent over-fitting. The hyper-parameters' values were determined via grid search.

## Results

### Exploration studies

This section details the results of exploration studies using the IXMAS dataset. The experiments included:

1. Performance comparison of LSTMRes with LSTMResKim [54], LSTM and Convolutional LSTM (ConvLSTM) [60].

2. Investigation on the impact of fine-tuning pre-trained CNNs with VGG-19 and VGG-16; other models, such as ResNet [52] and Inception [61] were not used because they impaired the recognition rate of the proposed model.

3. Evaluation of the multi-model approach and shared-weight technique.

4. Comparison of features fusion with score fusion using arithmetic and geometric means.

In every experiment, we used the most optimal structure for the succeeding experiment.

**LSTMRes vs LSTMResKim.** Fig 3 depicts the performances of LSTM, LSTMRes, and LSTMResKim on IXMAS based on training and validation errors. The results suggested that

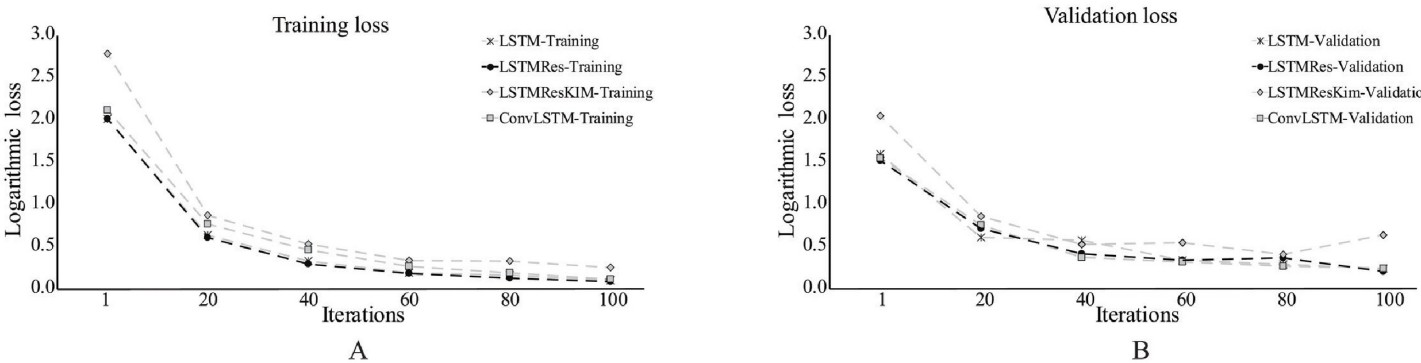

**Fig 3. Training and validation errors of LSTM, LSTMRes, LSTMResKim, and ConvLSTM.** From the start until the end of learning, LSTMResKim had lower performance than the others.

validation errors of the proposed model with the LSTMRes were lower than that with LSTMResKim. The training error with LSTMRes decreased steadily throughout the learning process. Instability, however, appeared in those of LSTMResKim after the 60th iteration.

In contrast, the LSTMRes exhibited slightly lower training and validation loss than LSTM and ConvLSTM. The performances of LSTM and LSTMRes were identical. These outcomes indicated that performing residual learning in the LSTM memory cell provides insignificant improvement with the model. In consideration of these results, we used LSTMRes used for the rest of the experiments reported here.

**Pre-trained CNNs.** VGG-16 and VGG-19 models trained with the ImageNet dataset were examined as CNNs blocks for the proposed model. We conducted three experiments using the intermediate `block4_pool` and the last `block5_pool` layers to find an appropriate pre-trained model and clarify the effect of fine-tuning. The first experiment used the intermediate layer as a feature extractor without fine-tuning the parameters, while the second applied it. The last experiment was conducted by fine-tuning the CNNs (from `block4_conv2` to `block5_conv3`).

Fig 4 illustrates the performance of the proposed model with different CNNs blocks. The results demonstrated three things. First, employing VGG-16 as a CNN block produced higher

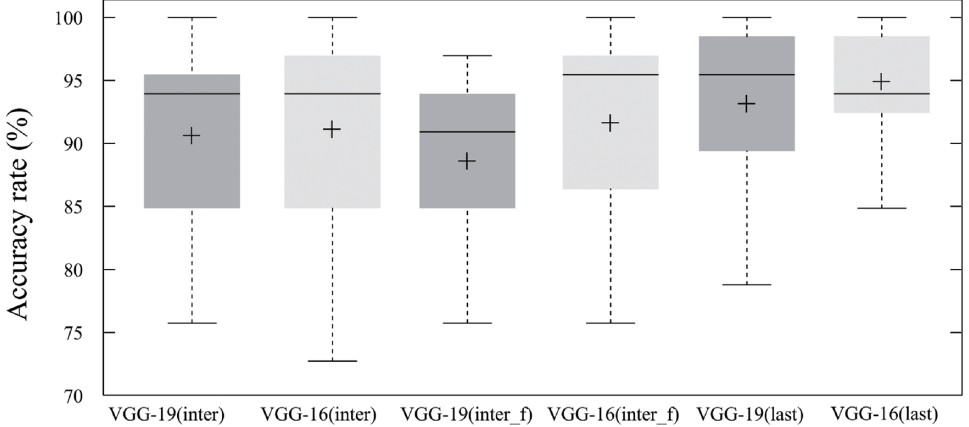

**Fig 4. Accuracy of the proposed model with different pre-trained CNNs.** Average recognition rates of the proposed model: VGG-19(intermediate): 90.40%; VGG-16(intermediate): 90.90%; VGG-19(fine-tuned intermediate): 88.38%; VGG-16(fine-tuned intermediate): 91.41%;VGG-19(last): 92.92%; VGG-16(last): 94.69%.

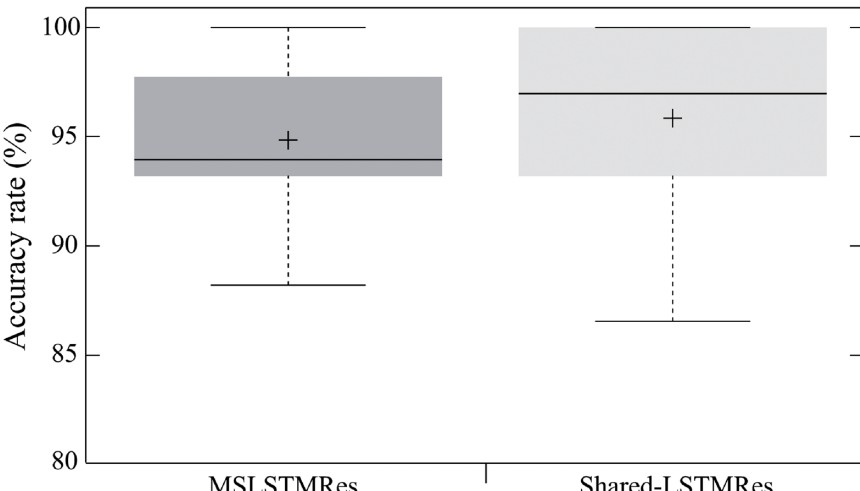

**Fig 5. Average accuracy rate of the proposed model.** Comparison between MSLSTMRes (94.69%) and MV-DNN (95.70%).

accuracy rate. The proposed model achieved an average increase in accuracy rate by 1.14 ±0.89% using either the intermediate or last-layer of VGG-16. Second, fine-tuning the last block of pre-trained CNNs improved insignificantly ($n = 396$, average $p > 0.05$) the proposed model's performance. Fine-tuning VGG-16 and VGG-19 improved respectively the model's accuracy rate by 4.29% and 2.52%. Third, fine-tuning of pre-trained CNNs parameters impaired the performance of the proposed model; its accuracy rate decreased by 2.02% with fine-tuning `block4` (intermediate) of VGG-19.

**Shared weight, no MSLSTMRes.** We previously found that MSLTMres yielded higher accuracy than the baseline model [23]. However, the recognition rate came at the expense of computational time and parameter numbers.

Given the benefits of shared-layer DNN in language modeling [55], we investigated related effects in multi-view action recognition, sharing the pre-trained VGG-16 and stacked LSTMRes of the proposed model across inputs from all cameras. Different attention layers were used for different views, and feature fusion was used to compute action probability.

The proposed model obtained a 1.01% ($n = 396$, $p = 0.617$) higher accuracy than with the use of MSLSTMRes (Fig 5). Shared-weight application also resulted in fewer parameters (the proposed model: 70,304,393, [23]: 351,323,711) and lower complexity with the proposed model, improving computation time.

**Score fusion.** This experiment compared the proposed approach's performance when using features and score fusions. Features fusion estimates action probability using a combination of the features from multiple cameras. Scores fusion, however, combined the prediction scores from multi-view inputs using the arithmetic or geometric mean.

This experiment used pre-trained VGG-16 and shared-weight LSTMRes as CNNs and RNNs layer for the proposed model, respectively. The models were trained with scenario II (Section Pre-processing and learning).

The proposed DNNs model exhibited an average accuracy of 97.22% with the use of arithmetic mean (Fig 6), which was 1.51% higher than using feature fusion. Scores fusion with the geometric mean created the opposite effect, decreasing the proposed model's accuracy rate by 1.77%. These results suggested the proposed DNNs model performed better when using the arithmetic mean as the score fusion.

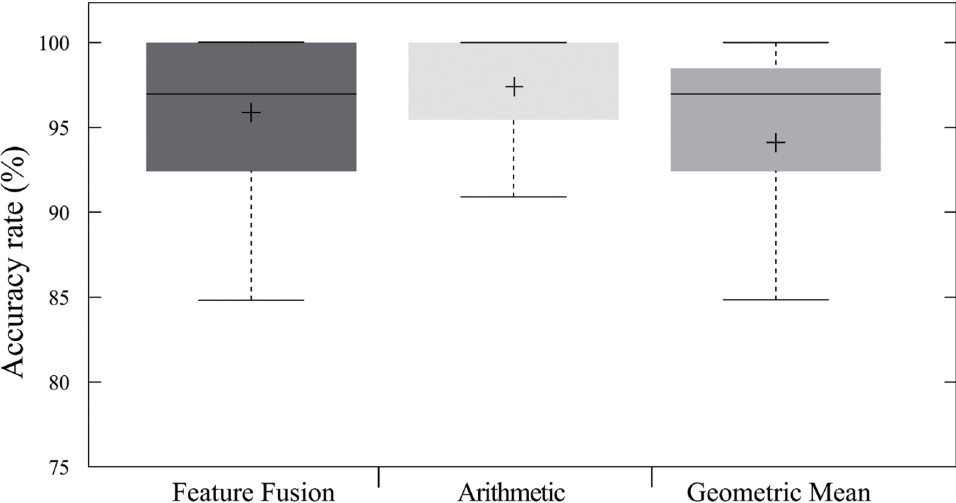

**Fig 6. Average accuracy rate of the proposed model using LSTMRes with feature-fusion and score-fusion techniques.** Accuracy of shared-weights LSTMRes with feature fusion (95.71%), and score fusion employing the arithmetic mean (97.22%) and geometric mean (93.93%).

**Final configuration.** The above exploratory studies showed that the employment of VGG-16 (block5_pool), shared-weight LSTMRes, and shared fusion with the arithmetic mean significantly improved ($n$ = 396, $p$ = 0.004) the performance of the proposed model by 7.07 ±14.03%, compared to the model in our previous work [23]. The highest improvement (4.29%) was observed with the use pre-trained VGG-16 as a CNN block and fine-tuning of its last layer (Table 2), while the lowest improvement (1.01%) was gained when replacing MSLSTMRes with shared LSTMRes. Although higher-resolution than in our previous research [23] was used here (128 x 128 vs 73 x 73 pixels), the recognition rate with the proposed model increased by 0.25%, which was insignificant, considering the small dataset.

Application of the optimized configuration also increased the accuracy of the proposed model in recognizing actions performed by hands (e.g., watching check, arms crossing, waving, and punching) (Fig 7). The highest improvement (25%) was achieved in the identification of "wave" action. We used the aforementioned new configuration for the next experiments: comparison of the proposed model with a single-input model and the state-of-the-art methods.

## Comparison with a single-input model

We performed an experiment on IXMAS with 13 actions to evaluate the proposed model's performance using single-view and multi-view inputs. Comparison of results (Fig 8)

**Table 2. Average accuracy gain (%)with new configuration.**

| Configuration | Accuracy | Improvement |
|---|---|---|
| Previous work [23] | 90.15 | - |
| 128 x 128 image | 90.40 | 0.25 |
| **Pre-trained VGG-16** | 94.69 | **4.29** |
| Shared-LSTMRes | 95.70 | 1.01 |
| Score fusion | 97.22 | 1.52 |
| **Total** | | 7.07 |

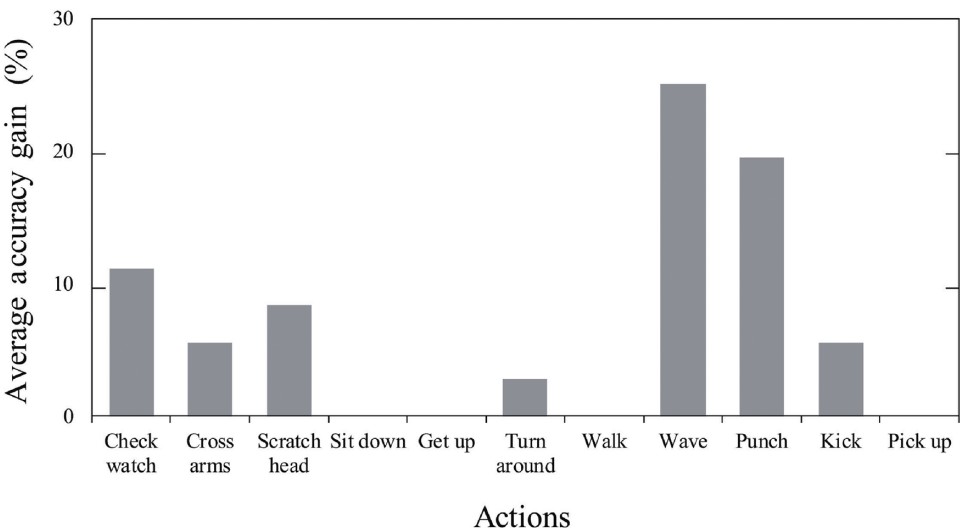

**Fig 7. Improvement of recognition rate with the new configuration.** There was no improvement in recognizing the-"sit down/get up/pick up"-actions, as perfect recognition rate was achieved by the model with the previous structure. The highest accuracy gain was in recognizing wave action (25%).

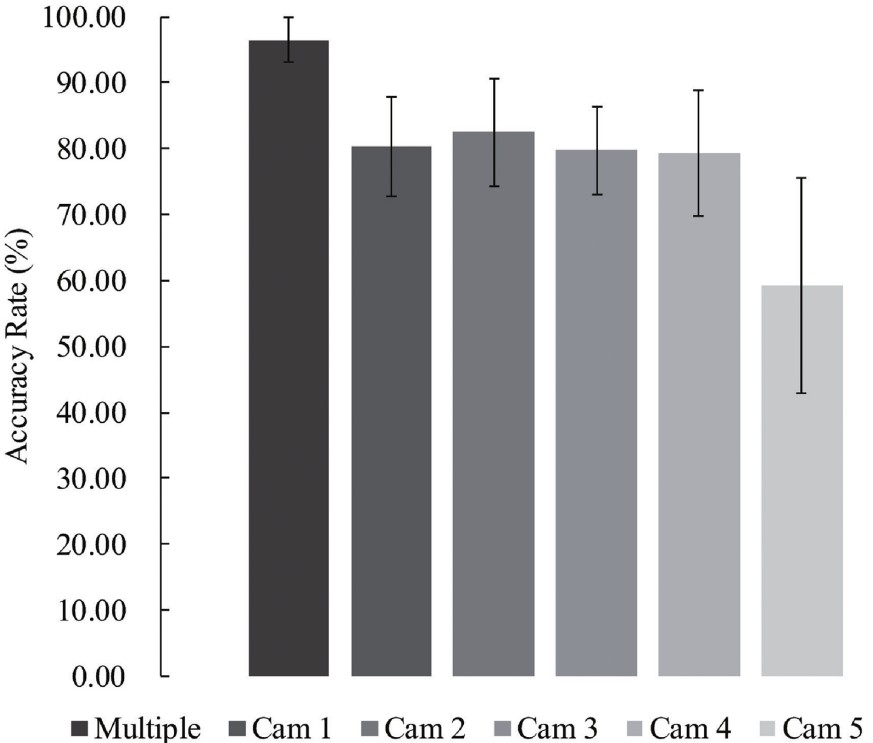

**Fig 8. Comparison between multi-view and single-view approaches.** Recognition rate of proposed model using multi-view inputs (96.37±3.39%) and single-view inputs from Cam 1 (80.34±7.57%), Cam 2 (82.48±8.10%), Cam 3 (79.70±6.66%), Cam 4 (79.27±9.56%), and Cam 5 (59.19±16.37%).

demonstrated that combining information from multi-view inputs produced a significant improvement in accuracy rate by 20.17±8.57% ($n = 396$, average $p < 0.05$). Compared to multi-view applications, the outcomes also indicated that the use of input from **Cam5** (top-view) produced a 37.18% lower recognition rate, while the employment of the other views yielded a 15.92±1.23% lower accuracy rate.

## Comparison with state-of-the-art methods

We compared the proposed model to state-of-the-art methods on IXMAS and i3DPost (Tables 3 and 4). Note that their results were not reproduced and the proposed model used 2D RGB images as inputs. Following the previous studies' experiment protocol, we evaluated the proposed model on the IXMAS dataset, with 11 subjects performing 10-action. We used data of all subjects in the evaluation of 13 actions on IXMAS and 10 and 12 actions on i3DPost. This experiment used learning scenario III (Sec. Pre-processing and learning).

**Table 3. Comparison for recognition (%) using the proposed model with state-of-the-art methods on IXMAS.**

| Method | Input | 11 Actions | 13 Actions |
|---|---|---|---|
| Holte *et al.* [9] | 4D | 100.00 | 100.00 |
| Turaga *et al.* [17] | 3D | 98.78 | - |
| Spurlock *et al.* [18] | Dynamic | 94.24 | - |
| Weinland *et al.* [22] | 3D | 93.30 | - |
| Pehlivan *et al.* [8] | 3D | 90.91 | 88.63 |
| Vitaladevuni [39] | 2D | 87.00 | - |
| Chaaraoui *et al.* [15] | 2D | 85.86 | - |
| Liu *et al.* [38] | 2D | 82.80 | - |
| Khan *et al.* [32] * | 3D | 99.60 | - |
| Gao *et al.* [21] * | 2D + optical flow | 99.60 | - |
| Purwanto *et al.* [34] * | 2D + optical flow | 97.22 | - |
| Gnouma *et al.* [33] * | 2D | 92.81 | - |
| **Proposed model** | 2D | **97.27** | **96.37** |

The "Input"-column indicates type of features used in the approaches. Khan *et al.* [32] utilized 50:50 training and test evaluation method, while Gnouma *et al.* [33] only evaluated their model with 10 actions.

* shows DNN based approach or the methods employing CNN based features.

**Table 4. Comparison for recognition (%) of proposed model with state-of-the-art methods on i3DPost.**

| Method | Input | 10 Actions | 12 Actions |
|---|---|---|---|
| Spurlock *et al.* [18] | Dynamic | 97.65 | - |
| Holte *et al.* [9] | 4D | 97.50 | - |
| Kose *et al.* [16] | 3D | 95.50 | - |
| Tran *et al.* [62] * | 3D | 96.70 | - |
| Mygdalis *et al.* [63] * | 3D | 95.51 | - |
| Angelini *et al.* [44] * | Skeleton | 99.47 | - |
| **Proposed model** | 2D | **93.75** | **96.87** |

The "Input"-column shows the type of features used in the methods. Evaluation was performed with the proposed model based on actions performed by one subject and two subjects ("12 Actions"-column). Mygdalis *et al.* [63] validated their model's performance using 3-fold cross-validation.

* shows DNN based approach or the methods employing DNN based features.

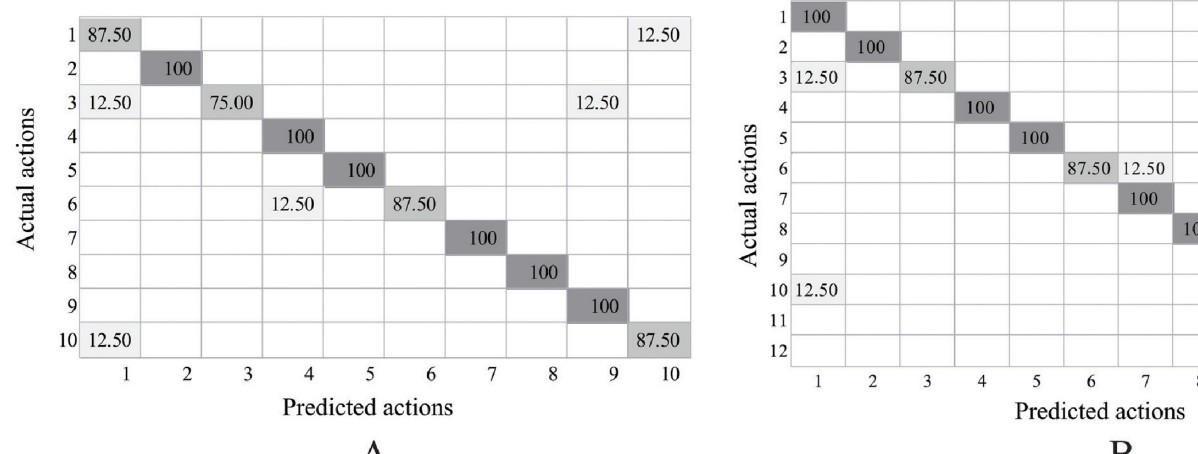

**Fig 9. Average accuracy rate of the proposed model on i3DPost.** The row and column represents action: walk(1), run(2), jump(3), bend(4), hand-wave(5), jump-in-place(6), sit-stand up(7), run-fall(8), walk-sit(9), run-jump-walk(10), handshake(11), pull(12). A and B illustrate experimental result on 10 and 12 actions, respectively.

In evaluation with IXMAS, the proposed model outperformed all 2D methods in recognizing 11 actions by 12.05% on average (Table 3). Performance was higher than methods employing 3D features representation [8, 22] but was slightly lower than the outcomes reported in [17]. The proposed model also got a 4.46% higher accuracy rate than other DNN models using 2D inputs and achieved competitive results to the models using 2D + optical flow inputs. However, the accuracy rate of the proposed model was 2.33% lower than that of the adaptive score fusion method.

In addition, the model produced a recognition rate of 96.37% in classifying 13 actions, outperforming Pehlivan *et al.* [8] with the use of 3D features. However, the proposed DNN model's recognition rate was still lower than 4D models [9].

The performance of the proposed DNN model in recognizing 10 actions on i3DPost was comparable to state-of-the-art methods (Table 4). The proposed model often misclassified actions with similar body configurations, such as jumping and bending, and exhibited confusion with differentiation of single and combined actions, such as "walking" and "running-jumping-walking" (Fig 9). The model achieved higher performance in classifying 10 actions and 2 interactions.

The proposed model obtained an average F1-score higher than 0.9 for all classes with all datasets (Table 5). The proposed model achieved the lowest F1-score when evaluated with 10-action on i3DPost and attained the highest F1-score on evaluation with 11-action on IXMAS.

**Table 5. Average F1-score of the proposed model with 11 and 13 actions of IXMAS, and 10 and 12 actions of i3DPost.**

|  | F1-Score (mean ± S.D.) |
|---|---|
| IXMAS-11 | 0.975 ± 0.026 |
| IXMAS-13 | 0.963 ± 0.025 |
| i3DPost-10 | 0.937 ± 0.062 |
| i3DPost-12 | 0.969 ± 0.038 |

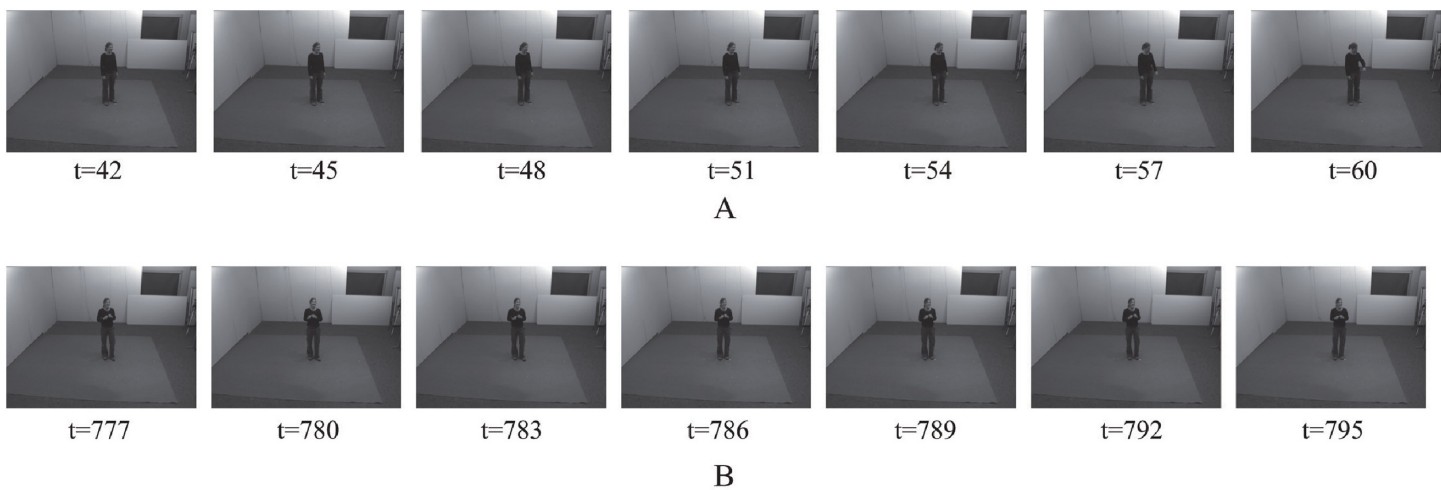

**Fig 10. Example of ambiguous-action clips.** A: sequence of images from early watch-checking action. B: sequence of ambiguous actions (transition from punching to kicking action).

## Online classification

In the online scenario, we did not segment individual action sequences based on the action labels, but used a sliding window to create clips from video content. The proposed model should determine early and ambiguous actions (Fig 10) from unfinished sequences of actions or transitions phases between actions. This study investigated how length of a clip affected the performance of the proposed model by setting the value of the sliding time window $t$ to 10, 20, 30, 40, and 50. The proposed model was trained based on learning scenario II (Section Pre-processing and learning) to estimate subjects' activity in each frame. The final prediction was the average probability scores over the sequence of images.

The experimental results (Fig 11) show that the highest accuracy and F1-score were attained with $t$ = 50. The accuracy and F1-score of the proposed model increased with longer sliding

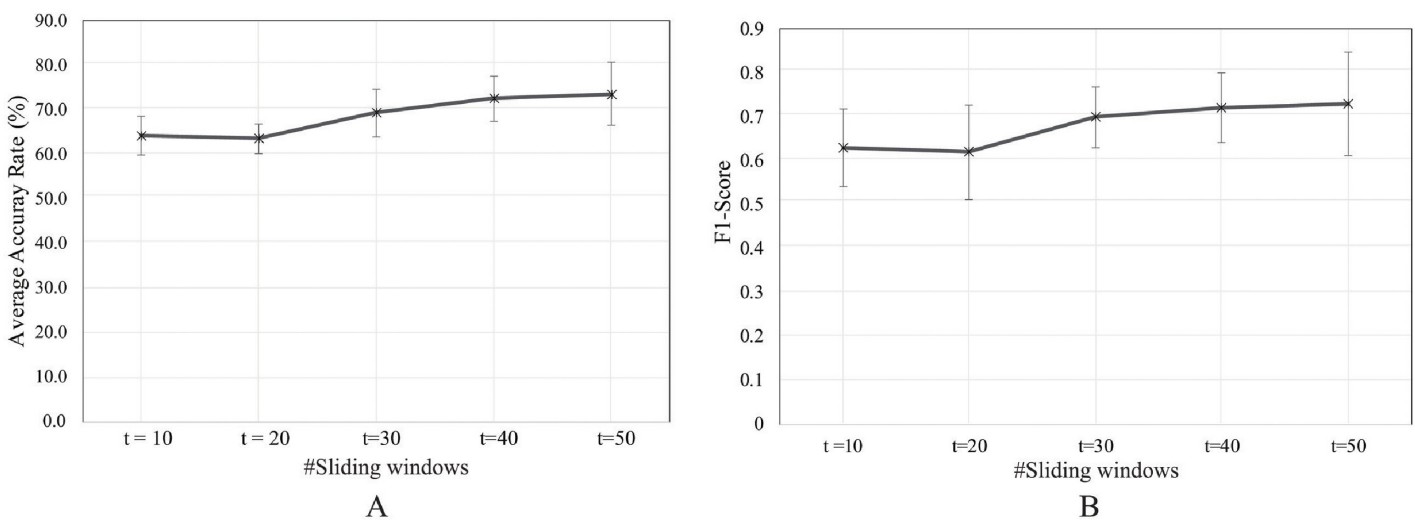

**Fig 11. Average accuracy rate of the proposed model in online classification.** A: accuracy of the proposed model with a varying number of sliding windows: t = 10 (64.24 ± 4.26%); t = 20 (63.55 ± 3.45%); t = 30 (69.36 ± 5.43%), t = 40 (72.60 ± 5.15%), and t = 50 (73.64 ± 7.15%). B: average F1-score of the proposed model with a varying number of sliding windows: t = 10 (0.63 ± 0.08); t = 20 (0.62 ± 0.10); t = 30 (0.7 ± 0.06%), t = 40 (0.72 ± 0.07%), and t = 50 (0.73 ± 0.11%).

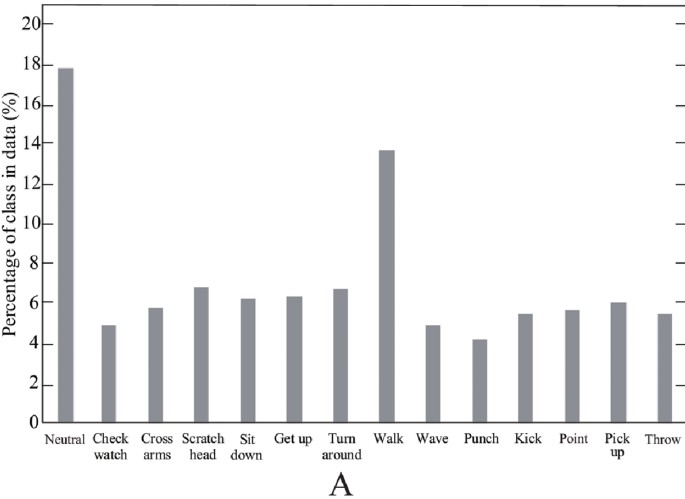
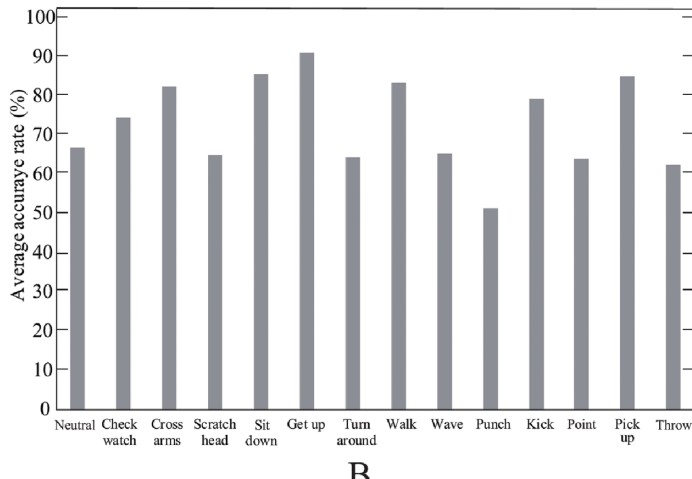

**Fig 12. Percentage labels in dataset and accuracy rate of the proposed model.** A: percentage of classes in IXMAS dataset segmented with *t* equaled to 50. B: accuracy of the proposed model for each class with t = 50.

window values. However, this did not represent a proportional correlation, as the recognition rate was 0.69% lower at t = 20 than at t = 10.

The imbalance dataset (Fig 12(A)) did not impair the overall performance of the proposed model: the proposed model achieved F1 scores higher than 0.6 in all scenarios. Besides, the proposed model classified sitting-down, getting-up, and picking-up actions with over 80% accuracy rate, even though the percentage of data based on such actions was lower than the others. However, the experimental results for t = 50 (Fig 12(B)) shows issues with the proposed model in differentiating actions performed only by hands (e.g., head-scratching, waving, punching, pointing, and throwing), the recognition rate at less than 70%.

## Discussion

When a subject performs activity in a dynamic environment, self and inter-object occlusion may occur. Multi-view human activity recognition helps to prevent a complete loss of information when occlusion appears in a single camera by providing information from other cameras [5, 11]. Previous findings have indicated that employing multiple view increased the recognition rate of human activity in a dynamic environment [6, 7].

This study presents a novel DNN model employing shared-weight and score fusion to classify human activity from multiple views. The experimental results suggested that the proposed model achieved optimal performance with VGG-16 as the pre-trained CNNs, shared-weight LSTMRes as RNNs layer, and average mean as score fusion. Exploratory studies showed that fine-tuning of pre-trained CNN parameters may not improve accuracy; fine-tuning `block4` of VGG-19 impaired the performance of the proposed model. We also found the model was better co-adapted with shallow pre-trained CNNs, as shown by improved performance for transfer learning with VGG-16. Those outcomes were attributed to VGG-16's output, which produced less domain-specific features than VGG-19.

Comparison between the proposed LSTMRes, and the method detailed by Kim [54] showed that shortcut connection between adjacent layer outputs of LSTM introduced instability, impairing performance. These results were consistent with those reported by Krueger *et al.* [64] and suggested an insignificant improvement in training and generalization performance with LSTM associated with residual learning in memory cells.

Second, the findings of this study suggested that even with a few training data, the proposed DNN model could attain competitive performance. The results showed that transfer-learning and score fusion with arithmetic mean improved the model's performance. Besides, compared to other DNN-based methods, the proposed model outperformed models using 2D modality [33] and achieved a lower accuracy rate than methods employing more complex modalities, such as optical flow [21, 34] and skeleton data [44].

Another major finding of this study was that the proposed model required a longer sliding window to attain optimal performance. That contradicts the results of previous work [65] that found a brief sequence was sufficient for the evaluation of basic human actions. One interpretation of these findings is that the applicable number of frames to recognize human activities may differ from case to case; we used different datasets from Schindler *et al*. [65]. In our experiment, short sequences resulted in ambiguous clips, impairing the performance of the proposed model.

As seen in previous works [1, 6], this study found that combining information from multiple views resulted in a higher accuracy rate of the proposed model in MVHAR. This suggested that additional information from another view mitigated information loss caused by occlusion. The results also suggested that the proposed model could filter out uninformative features, since the recognition rate did not decline when input from Cam5 was combined with other views; using single-view input from Cam5 resulted in impaired performance of the proposed model.

Despite promising results achieved with the proposed model compared to state-of-the-art methods, this study had several limitations. First, the model did not get satisfying results evaluated for online scenario requiring classification of sequences of ambiguous action. Second, the study only evaluated the proposed model's performance with two benchmark datasets, which comprised less than 15 subjects performing basic activities. Hence, the experimental results remain preliminary. Last, only self-occlusion was observed in the datasets. Accordingly, the proposed model's performance for mutual occlusion is unclear. Further study is needed to evaluate and improve model performance in an online scenario involving more subjects. It also would be of interest to consider evaluating the model with more benchmark datasets comprising complex activities performed in various situations, such as CASIA [66], UCF101 [67], MOD20 [68], and HMDB51 [69] datasets.

## Author Contributions

**Conceptualization:** Prasetia Utama Putra, Keisuke Shima.

**Funding acquisition:** Keisuke Shima, Koji Shimatani.

**Methodology:** Prasetia Utama Putra, Keisuke Shima.

**Supervision:** Keisuke Shima, Koji Shimatani.

**Validation:** Prasetia Utama Putra.

**Writing – original draft:** Prasetia Utama Putra.

**Writing – review & editing:** Prasetia Utama Putra, Keisuke Shima.

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
