## [Decision Letter · Decision Letter 0]

2 Nov 2021

PONE-D-21-30206A Deep Neural Network Model for Multi-View Human Activity RecognitionPLOS ONE

Dear Dr. Putra,

Thank you for submitting your manuscript to PLOS ONE. After careful consideration, we feel that it has merit but does not fully meet PLOS ONE’s publication criteria as it currently stands. Therefore, we invite you to submit a revised version of the manuscript that addresses the points raised during the review process.

Please, see your comments below and provide a point-by-point response.

We look forward to receiving your revised manuscript.

Kind regards,

Antonio Agudo

Academic Editor

PLOS ONE

Journal Requirements:

[NO authors have competing interests]. 

5. Please ensure that you refer to Figure 8 in your text as, if accepted, production will need this reference to link the reader to the figure.

Additional Editor Comments:

Dear authors,

both reviewers agree most of the comments were addressed properly, so thanks for that. However, some points need to be improved before proceeding, especially the questions provided by R1. Please, consider all the comments in your response point by point. In addition to that, as it was reported by R2, the paper could be rewritten in some sections to improve English style.

Best

Reviewers' comments:

Reviewer's Responses to Questions

**Comments to the Author**

1. Is the manuscript technically sound, and do the data support the conclusions?

Reviewer #1: Yes

Reviewer #2: Yes

2. Has the statistical analysis been performed appropriately and rigorously? 

Reviewer #1: No

Reviewer #2: Yes

3. Have the authors made all data underlying the findings in their manuscript fully available?

Reviewer #1: Yes

Reviewer #2: Yes

4. Is the manuscript presented in an intelligible fashion and written in standard English?

Reviewer #1: Yes

Reviewer #2: Yes

5. Review Comments to the Author

Reviewer #1: Authors should address the following minor issues:

1) In the introduction section, it is essential to discuss the applications of single view and multi-view in the revised version.

2) Discuss the purpose of multi-view and also describe its importance with related references. it is missing in the revised version.

3) The problem statement should be modified like what are the problems in single view and what is the need of multi-view?

4) The last contribution is just an experiment?

5) What are the parameters of LSTM? Have you make any changes?

6) What are the parameters of CNN? which layer is selected for feature mapping?

7) I am worried that why you select only two datasets? why not select UCF101 and HMDB51? Add this discussion in the future work.

8) The related work should be refine further and add the latest articles.. The following articles can be added.

- A Fused Heterogeneous Deep Neural Network and Robust Feature Selection Framework for Human Actions Recognition

- Real-time Violent Action Recognition Using Key Frames Extraction and Deep Learning

- Multi-Layered Deep Learning Features Fusion for Human Action Recognition

- Video Analytics Framework for Human Action Recognition

Reviewer #2: All the comments to the authors were satisfactorily addressed. The related work section was improved, in the experiments section the authors model is now evaluated in a single-view scenario. The English is still not perfect.

6. PLOS authors have the option to publish the peer review history of their article (what does this mean?). If published, this will include your full peer review and any attached files.

Reviewer #1: No

Reviewer #2: No

---

## [Author Response · Author response to Decision Letter 0]

13 Dec 2021

We would like to thank the reviewers for their comments. We write the responses to the reviewers’ specific concerns.

---

## [Editor Report · Decision Letter 1]

19 Dec 2021

A Deep Neural Network Model for Multi-View Human Activity Recognition

PONE-D-21-30206R1

Dear Dr. Putra,

We’re pleased to inform you that your manuscript has been judged scientifically suitable for publication and will be formally accepted for publication once it meets all outstanding technical requirements.

Kind regards,

Antonio Agudo

Academic Editor

PLOS ONE

Additional Editor Comments (optional):

All the comments were addressed properly. The paper is ready for publication. Best
---

## [Editor Report · Acceptance letter]

30 Dec 2021

PONE-D-21-30206R1 

A Deep Neural Network Model for Multi-View Human
Activity Recognition 

Dear Dr. Putra:

I'm pleased to inform you that your manuscript has been deemed suitable for publication in PLOS ONE. Congratulations! Your manuscript is now with our production department. 

Kind regards, 

on behalf of

Dr. Antonio Agudo 

Academic Editor

PLOS ONE